# Synthesis of Flavonol-Bearing Probes for Chemoproteomic and Bioinformatic Analyses of Asteraceae Petals in Search of Novel Flavonoid Enzymes

**DOI:** 10.3390/ijms24119724

**Published:** 2023-06-03

**Authors:** Karl Kempf, Oxana Kempf, Yoan Capello, Christian Molitor, Claire Lescoat, Rana Melhem, Stéphane Chaignepain, Elisabeth Génot, Alexis Groppi, Macha Nikolski, Heidi Halbwirth, Denis Deffieux, Stéphane Quideau

**Affiliations:** 1ISM (CNRS-UMR 5255), University of Bordeaux, 33405 Talence CEDEX, France; 2Institute of Chemical, Environmental & Bioscience Engineering, Technische Universität Wien, 1060 Vienna, Austria; 3Centre de Bioinformatique de Bordeaux (CBiB), University of Bordeaux, 33076 Bordeaux CEDEX, France; 4Centre de Recherche Cardio-Thoracique de Bordeaux (INSERM U1045), University of Bordeaux, 33607 Pessac CEDEX, France; 5CBMN (CNRS-UMR 5248), Centre de Génomique Fonctionnelle de Bordeaux, University of Bordeaux, 33076 Bordeaux CEDEX, France; 6IBGC (CNRS-UMR 5095), University of Bordeaux, 33077 Bordeaux CEDEX, France; 7Institut Universitaire de France, 75231 Paris CEDEX 05, France

**Keywords:** plant polyphenols, Asteraceae UV-honey guides, higher hydroxylated flavonols, flavonoid biosynthesis, chemoproteomics, bioinformatics

## Abstract

This study aimed at searching for the enzymes that are responsible for the higher hydroxylation of flavonols serving as UV-honey guides for pollinating insects on the petals of Asteraceae flowers. To achieve this aim, an affinity-based chemical proteomic approach was developed by relying on the use of quercetin-bearing biotinylated probes, which were thus designed and synthesized to selectively and covalently capture relevant flavonoid enzymes. Proteomic and bioinformatic analyses of proteins captured from petal microsomes of two Asteraceae species (*Rudbeckia hirta* and *Tagetes erecta*) revealed the presence of two flavonol 6-hydroxylases and several additional not fully characterized proteins as candidates for the identification of novel flavonol 8-hydroxylases, as well as relevant flavonol methyl- and glycosyltransferases. Generally speaking, this substrate-based proteome profiling methodology constitutes a powerful tool for the search for unknown (flavonoid) enzymes in plant protein extracts.

## 1. Introduction

Flavonols are polyphenolic secondary metabolites that fulfill important physiological functions related to plant fertility, growth, and defense [1,2]. These flavonoids are 3-hydroxy-2-phenylchromen-4-ones that are commonly hydroxylated (or methoxylated) at positions 5 and 7 (ring A) and at positions 3′ and/or 4′ (ring B), such as quercetin (**1**, Figure 1). In flowers of Asteraceae, flavonols bearing an additional hydroxyl group at positions 6 or 8 (ring A) contribute to petal UV-absorbing pigmentation patterns, which play a crucial role in attracting pollinating insects [2,3,4]. The 6-hydroxyquercetin (**2a**, quercetagetin) and the 8-hydroxyquercetin (**2b**, gossypetin), as well as their methylated and/or glycosylated derivatives, are much higher hydroxylated flavonols, thus serving as UV-honey guides (Figure 1) [5].

Understanding the biogenesis of these special flavonols found in Asteraceae species requires the identification of specific enzymes involved in the incorporation of an extra 6- or 8-hydroxyl group onto the quercetin ring A and its subsequent methylation or glycosylation. Biochemical studies on Asteraceae species demonstrated the presence of a cytochrome P450-dependent flavonol 6-hydroxylase (F6H) in enzyme preparations from the petals of *Tagetes patula*, *Tagetes erecta* [6], and *Rudbeckia hirta* [5]. The sequence of *Rudbeckia hirta* F6H (NCBI MT875175) was recently identified by homology cloning. In addition, the presence of an NADPH/FAD-dependent flavonol 8-hydroxylase (F8H) has been detected in the petals of *Chrysanthemum segetum* [7]. In species of other plant families, another type of F6H, a 2-oxoglutarate-dependent dioxygenase, was identified and found to be responsible for the 6-hydroxylation of methylated flavonols in the leaves of *Chrysosplenium americanum* (Saxifragaceae) [8,9]. A cytochrome P450-dependent flavanone 6-hydroxylase was also identified and reported to be involved in the biosynthesis of 6-hydroxyisoflavones in soybean (Fabaceae) [10]. More recently, an NADPH/FAD-dependent flavonoid 8-hydroxylase capable of synthesizing gossypetin (**2b**) as well as 8-hydroxykaempferol was identified from *Lotus Japonicus* (Fabaceae) [11]. However, in the case of Asteraceae species, neither the F8H nor the enzymes responsible for the methylation and/or glycosylation of **2a** and **2b** have yet been fully identified. Moreover, the genes coding for these functional proteins are still unknown, which has so far prevented any homology-based cloning approach from their full identification. 

## 2. Results and Discussion

This long-standing and still challenging situation regarding the identification of these flavonol hydroxylases in Asteraceae species prompted us to engage in an interdisciplinary dual approach in which affinity-based chemoproteomic profiling [12,13,14] complements our work on classical homology-based gene cloning. The aim of our affinity-based chemoproteomic approach is to identify relevant functional proteins through their (covalent) capture by chemical probes equipped with the main F6H and F8H substrates, quercetin (**1**). Herein, we describe the chemical synthesis of these probes, their implementation for the capture of Asteraceae proteins, and the results of our proteomic and bioinformatic analyses.

### 2.1. Chemical Synthesis of Quercetin-Bearing Probes

The design of those quercetin-bearing probes was inspired by our previous syntheses of other flavonoid-bearing probes (Figure 2) [15,16]. Quercetin (**1**) was first benzylated by taking advantage of the reaction temperature to generate either the perbenzylated quercetin **3a** at 70 °C or the tetrabenzylated quercetin **3b** [17] at room temperature, the milder conditions under which benzylation of the 5-hydroxy group (ring A) was prevented were probably because of its resisting H-bonding with the 4-ketone function (ring C) (Figure 2). Both compounds were then regioselectively iodinated using N-iodosuccinimide (NIS, 1.25 equiv.). The iodination of **3a** required the addition of silver(I) triflimide as a catalyst (0.2 equiv.) to activate NIS [18] and occurred at the most nucleophilic position 8 (ring A). The iodination of **3b** was directed by the 5-hydroxy group and selectively occurred at position 6 (ring A) without the addition of a silver(I) catalyst [19]. The resulting iodoarenes **4a** and **4b** then enabled us to install a carbonylated three-carbon tether at either iodinated position through Heck reactions using ethyl acrylate (10 equiv.) and palladium(II) acetate as catalysts (0.2 equiv.) [20]. Hydrogenolysis of the benzyl groups and selective hydrogenation of the olefinic bond of the resulting unsaturated esters **5a** and **5b** [21], followed by intramolecular transesterification at high temperature, quantitatively led to the formation of the lactones **6a** and **6b**. These lactones served as electrophiles in acyl substitution reactions with the previously described amino-polyethylene glycol (PEG) biotinylated derivative **7** [15,16] to form the two desired quercetin-bearing probes **Q8** and **Q6** (Figure 2). The structure of these probes leaves positions 6 or 8 (ring A) free to allow for recognition by the flavonol hydroxylases being sought. All phenolic hydroxyl groups were also left free since they are usually essential for recognition by and binding to proteins. The catechol B rings served to promote the covalent capture of interacting proteins through their oxidation into electrophilic ortho-quinones using an oxidant such as sodium periodate (NaIO_4_) [15]. The biotin unit served to retrieve probe-interacting proteins, covalently or not, from Asteraceae petal protein extracts with the help of streptavidin-coated magnetic beads [15]. An analogous probe **K8** (Figure 2) bearing the flavonol kaempferol, which features a monohydroxylated (not catechol) ring B but whose 5,7-dihydroxylated ring A can also be enzymatically further hydroxylated, was similarly synthesized [22] (see the Appendix A for details) and served as a comparative tool in our protein capture protocol.

### 2.2. Affinity-Based (Covalent) Capture of Asteraceae Proteins

Microsomal proteins were obtained from the petals of two Asteraceae species, namely *Rudbeckia hirta* and *Tagetes erecta* (see Section 3). These extracts were then submitted to our protein capture protocol [15], which was further optimized and adapted to these plant proteomic materials. Briefly, each of the two proteomes was incubated with each of the three probes (i.e., **Q8**, **Q6,** and **K8**), treated with NaIO_4_ (method **O**) or not (method **N**), and the resulting mixtures were incubated with streptavidin-coated magnetic beads to pull down proteins interacting with the biotinylated probes. The minimal quantity of NaIO_4_ required for converting the catechol units of probes **Q8** and **Q6** into *ortho*-quinones was carefully determined through a newly developed in situ redox titration method based on the use of 2,6-dichlorophenolindophenol (DCPIP) [23,24] (see the Appendix A, for details). This precaution was taken in order to avoid or at least minimize the oxidative cleavage of 2-amino alcohols or 1,2-diols in *N*-terminal serine or threonine residues or in carbohydrate moieties of glycosylated proteins, which may cause undesired protein cross-linking events [25]. Thus, the quantity of NaIO_4_ and the time of the oxidation reaction, as well as the composition of the buffered solutions used, were better optimized for proper activation of the probes and capture of the interacting proteins (see the Section 3 and the Appendix A for details).

Proteins captured by the probes and enriched from the initial proteomes were then submitted to shotgun proteomics using quantitative label-free LC-MS/MS techniques (all assays were performed in triplicate). Further identification of proteins then relied on the interrogation of several Asteraceae databases, namely the proteome data of *Helianthus annuus* [26], the transcriptome data of *Tagetes erecta* [27], and the transcriptome data we collected for *Rudbeckia hirta* (NCBI BioProject PRJNA631685).

### 2.3. Proteomic and Bioinformatic Analyses of Proteins Captured from Rudbeckia Hirta

This chemical proteomic analysis of two different Asteraceae species using three different molecular probes through two different methods (Figure 3A) generated large and complex data sets. Based on these data sets, we first aimed to verify if known proteins captured by our flavonol-bearing probes were indeed involved in flavonoid biosynthesis. The first set of data we analyzed were those resulting from capture experiments performed on *Rudbeckia hirta* microsomal petal proteins by using the quercetin-bearing probes **Q6** and **Q8** and the kaempferol-bearing probe **K8**. Out of 1931 proteins identified in this microsomal extract through interrogation of the *Helianthus annuus* total proteome [26], 836 proteins were detected in the capture sample using probe **Q8** through method **O**. A bioinformatic processing of these capture results was carried out to select the most reliable data (see the Appendix A, for details), conserving only those proteins that had a sufficient reproducibility of capture as calculated from the three experimental replicates. From this selection of 768 proteins, the captures of 368 proteins were rated as statistically highly significant, i.e., not detected in the probe-less negative control or highly significantly enriched in the capture sample (Figure 3B). Among these proteins, 3 were identified as being involved in flavonoid biosynthesis, namely a chalcone-flavanone isomerase (CHI), a flavonoid 3’-hydroxylase (F3′H), and most satisfyingly, the *Rudbeckia* flavonol 6-hydroxylase (F6H). The sequence of this F6H from *Rudbeckia hirta* that we recently identified by homology cloning (NCBI MT875175) was manually added to the *Helianthus* database used for the interrogation of our capture samples. The same flavonoid-specific enzymes, including F6H, were also among the 355 highly significant proteins captured by probe **Q6** (Figure 3B). In fact, 345 of those proteins were captured by both probes **Q6** and **Q8** (Figure 3C, see Venn diagram I). These observations would indicate that both regioisomeric probes interact with proteins, recognizing mainly their flavonol core independently from their site of attachment to the rest of the probe structures.

The comparison of captures by probes **Q8** or **Q6** and **K8** revealed that probe **K8**, having a non-catechol ring B not directly convertible into an *ortho*-quinone unit by a simple NaIO_4_-mediated dehydrogenation, is a much less efficient capture tool. For example, only 121 proteins were captured by probe **K8**, and 118 of those were also captured by probe **Q8** (Figure 3C, see Venn diagram II). In addition, a relatively high number of proteins (402, see Figure 3B) were also captured through method **N** (i.e., without NaIO_4_-promoted activation of probe **Q8**). Out of the 368 proteins retrieved through method **O**, 304 were also identified through method **N** (Figure 3C, see Venn diagram III), including the same aforementioned flavonoid enzymes (i.e., F6H, F3′H, and CHI) and two polyphenol oxidases (PPOs). These captures through both methods **N** and **O** would either imply that probe **Q8** has a sufficiently high (and resistant) affinity for these proteins to be retrieved without any formation of covalent linkages or that such covalent linkages would be formed through an oxygen-dependent catechol oxidation process (i.e., autoxidation) during the probe/protein incubation step. Upon interrogation of our *Rudbeckia* transcriptomic data (NCBI BioProject PRJNA631685), a much higher number of proteins was detected in the microsomal extract (i.e., 3332 instead of 1931), as well as in capture samples (Figure 3B). One of the reasons for these higher numbers is certainly due to the non-fully curated nature of the transcriptome database, which contains several redundancies (i.e., identical proteins corresponding to different entries in the database). Nevertheless, several hydroxylases, such as a salicylate 5-hydroxylase (S5H), two cinnamate 4-monooxygenases (C4H and C4HL), and also a “putative” flavanone 3-dioxygenase (FHT), were additionally identified from this *Rudbeckia* transcriptomic data interrogation. Of note is that this putative FHT, as well as a chalcone synthase (CHS) and one of the PPOs annotated with the annotation “pigment biosynthetic process”, were captured at more significant levels through method **N**. These first comparative results of our different capture assays are summarized in the heatmap displayed in Figure 4.

### 2.4. Proteomic-Based Bioinformatic Search of Novel Flavonoid Enzymes in Rudbeckia Hirta and Tagetes Erecta

The above initial analysis of our *Rudbeckia* proteomic data thus clearly confirmed the functional value of our quercetin-bearing probes **Q6** and **Q8**, since they are capable of capturing flavonoid-specific enzymes. Most significantly, the capture of the F6H by both probes boded well for the search for other quercetin-modifying enzymes, such as the F8H, in Asteraceae protein extracts. The next step of our chemical proteomic data analysis was thus to shortlist certain uncharacterized captured proteins as possible novel enzymes involved in flavonoid biosynthesis, such as the F8H, as well as methyl- and/or glycosyltransferases acting on the higher hydroxylated flavonols **2a** and **2b**. Potentially relevant uncharacterized or not fully characterized (i.e., not yet soundly annotated in databases) proteins among the highly significant proteins captured through our methods are listed in the heatmap displayed in Figure 5. Most proteins, even those listed as uncharacterized proteins (u. prot.), are annotated with predicted functions based on sequence similarities [i.e., gene ontology (GO) and InterPro classification (IPR)]. We notably looked for keywords that could help us uncover the identity of uncharacterized proteins with functions potentially relevant to the activity of oxidases (Figure 5A), methyltransferases (Figure 5B), or glycosyltransferases (Figure 5C).

The same chemical proteomic analyses were also performed on *Tagetes erecta* microsomes using probe **Q8** through method **O** (Figure 5), and the captured proteins were identified through interrogations of the *Helianthus* proteome and *Tagetes* transcriptome data [26,27]. Although a much lower number of proteins were highly significantly captured from these *Tagetes* protein samples (i.e., out of 2777 proteins in the total proteome, only 130 were detected in the capture samples), the captures of 44 proteins were rated as highly significant (see the Appendix A), and 5 proteins with relevance to our search were identified (Figure 5). The coherence of all assays was assessed by referring to the *Helianthus* homologues of the captured proteins and was confirmed by the capture of those five relevant proteins from both proteomes (Figure 5). In particular, the “probable” methyl transferase 3 (PMT3) and again the F6H were thus captured from both *Tagetes* and *Rudbeckia* petals. Interestingly, both F6H sequences identified from *Rudbeckia* and *Tagetes* correspond to the *Helianthus* homologue XP_022000023, which is named “probable (*S*)-*N*-methylcoclaurine 3’-hydroxylase isozyme 2 isoform X1” (first entry in Figure 5). The *Tagetes* F6H (NCBI MT882687) was also identified by homology cloning, and its sequence was thus also used for interrogation of our capture data. Two additional relevant yet uncharacterized oxidoreductases and one uncharacterized *O*-linked *N*-acetylglucosamine (GlcNAc) transferase were also highly significantly captured from *Tagetes* petals (Figure 5). Their presence in the *Rudbeckia* protein extract was only confirmed by low fold changes observed mainly using method **O** and probes **Q6**, **Q8**, and **K8**. These proteins constitute interesting candidates for the identification of novel enzymes, including the F8H, for the flavonoid metabolism in Asteraceae species.

A quick look at the heatmap displayed in Figure 5 again reveals a similarly high level of protein capture by both isomeric quercetin-bearing probes **Q6** and **Q8** and a significantly lower level of capture using the kaempferol-bearing probe **K8**. However, we basically have no indication as to whether these captured proteins preferentially act on a given flavonoid or another. Such a structural specificity issue can only be assessed on the basis of the five proteins that were selectively captured by only one of the two probes, **Q6** or **Q8** (these proteins are highlighted by white frames in Figure 5). We thus continued to search for the means to reveal the relative specific affinity of these two quercetin-bearing probes for proteins captured by both probes. In general, a higher affinity for a given protein leads to its higher abundance in the capture sample, which is usually determined relative to the abundance of the same protein in the negative probe-less control sample. However, the most highly significant captured proteins were not detected in negative control samples. Therefore, the level of selectivity of the two probes **Q6** and **Q8** was assessed by direct comparison of the protein abundances in the capture samples. Hence, the fold change of a protein in the **Q8** capture sample is determined by its abundance in the **Q8** sample relative to that in the **Q6** capture sample, and vice versa. 

The results of this comparative analysis are depicted in a volcano plot (Figure 6), in which the ratio r of the relative abundance is plotted against the *p*-value [i.e., X-axis = lg2(r) for Q8 vs. Q6 and –lg2(r) for Q6 vs. Q8; Y-axis = –log(p)]. Only 2 out of the 24 proteins that are displayed in Figure 6 exhibited a significant fold change upon using probe **Q6** *versus* **Q8**, a translation initiation factor, and an aminoacyl-tRNA synthetase, thus both are involved in protein synthesis but not in flavonoid metabolism. This result means that probe **Q6**, even though it captures essentially the same proteins as probe **Q8** (see Figure 3C, Venn diagram I), binds to most of them with a significantly lower affinity. On the other hand, among the substantial number of proteins with a significant fold change upon using probe **Q8** versus **Q6** are several potentially relevant enzymes. The protein showing the highest fold change upon using probe **Q8** is a peroxidase 12-like protein. Further significantly enriched proteins include 2 out of the 5 probable methyltransferases (PMT26 and PMT11, see Figure 5B) and a known phenolic methyl transferase (caffeic acid 3-*O*-methyltransferase). The uncharacterized protein (LOC110904084) is also a methyl transferase annotated as putatively acting on the C5 position of the base uracil (IPR010280). However, it should also be considered a candidate for further identification of relevant proteins since our experimental data thus indicate a selective flavonol affinity (see Figure 5B and Figure 6).

### 2.5. BlastP-Based Bioinformatic Search of Rudbeckia Flavonol Hydroxylases

Finally, with the *Rudbeckia* F6H sequence in hand, we also performed a blastP search [28] on the *Rudbeckia* transcriptome data (NCBI BioProject PRJNA631685), which yielded 30 sequences producing significant alignments. From the five hits with the highest similarity scores (>500, see Figure 7A), two sequences are those of proteins that were highly significantly captured by the quercetin-bearing probes **Q8** and **Q6**, namely again the F6H itself, which is annotated here as the *Helianthus* homologue referred to as “(*S*)-*N*-methylcoclaurine 3’-hydroxylase” [TRINITY_DN77218_c0_g1_i2.p1], and another protein referred to as “geraniol-8-hydroxylase” [TRINITY_DN75109_c0_g1_i2.p1]. These blastP results indicate that our probes bind to and capture certain folded proteins having primary structures (i.e., sequences) similar to those of other proteins with no affinity for our probes, hence also confirming the specificity of the observed probe-protein interactions. Among the other 25 sequences with a much lower F6H similarity score, only one protein was significantly captured by probes **Q8** and **Q6**. This protein corresponds to the *Helianthus* homologue referred to as “cytochrome P450 Tp4149” [TRINITY_DN69456_c0_g1_i1.p1]. Both proteins with hydroxylase activity constitute candidates for the identification of F8H in the Asteraceae species.

Thus, candidates for an F8H in *Rudbeckia* can either be identified by their similarity to the *Rudbeckia* F6H (Figure 7A) or to other known F8H sequences. In the absence of known relevant proteins in Asteraceae species, known flavonol hydroxylase sequences from species of other plant families (i.e., *Lotus japonicus* (Fabaceae) [11] and *Scutellaria baicalensis* (Lamiaceae) [29]) were integrated into a BlastP search on the *Rudbeckia* transcriptome data. None of the proteins captured by our probes were similar to the *Lotus* F8H. However, out of the 35 sequences producing significant alignments with *Scutellaria* F6H and F8H, one protein was captured by our probes **Q6** and **Q8** (Figure 7B). Interestingly, this protein is the same cytochrome P450 oxygenase that had already been identified with a moderate similarity score against the *Rudbeckia* F6H sequence (see Figure 7A, last entry). Overall, this combined protein capture/BlastP approach thus constitutes an effective tool for shortlisting candidates for the search of relevant flavonoid enzymes, here highlighting two proteins (a “geraniol-8-hydroxylase” and the cytochrome P450 Tp4149) as candidates for cloning and evaluation of their F8H activity.

## 3. Materials and Methods

### 3.1. Microsomal Protein Extraction

For the microsomal protein preparations, 5 g of petals and 2.5 g of Polyclar AT, 2.5 g of quartz sand, and 30 g of a buffer solution (0.1 M KH_2_PO_4_, 10% saccharose, 0.4% sodium ascorbate, pH 7.5) were homogenized in a mortar, kept on ice for 5 min, and centrifuged at 8000× *g* for 20 min at 4 °C. The supernatants were filtered through quartz wool, and the pellets were discarded. The resulting filtrates were subjected to ultracentrifugation at 36,000× *g* for 60 min at 4 °C. The supernatants were discarded, and the pellets were washed with 0.1 M TRIS buffer at pH 6.5. The pellets were then resuspended, homogenized at 4 °C, and shock-frozen in liquid nitrogen. The samples were stored at −80 °C until further use.

### 3.2. Affinity-Based (Covalent) Protein Capture Assays

*Source of proteins*: Suspension of microsomes with petal proteins from *Rudbeckia hirta* (~1.5 mg/mL) and *Tagetes erecta* (~2–2.5 mg/mL) in buffer (0,1 M TRIS/HCl, 20% saccharose, 0.4% ascorbate).

*Probe excess:* The NaIO_4_-mediated oxidative covalent capture of proteins proceeds with high efficiency with two equivalents of the probe relative to the approximate binding capacity of the streptavidin-coated magnetic beads. This minimal excess of the probe saves additional steps usually required for the separation of the proteins from the non-reacted excess of probes [30].

*Type of buffer:* The ideal buffer for the NaIO_4_-mediated oxidation step is PBS because it is redox-neutral. Organic alcohols such as TRIS [tris(hydroxymethyl)aminomethane] or 1,2-diols should be avoided since aldehydes possibly resulting from oxidative cleavage reactions may cause unwanted covalent modifications of proteins. The 1,2-diol dithiothreitol (DTT) should thus not be used for the same reason and also because its thiol groups are known to form covalent adducts with probe-derived *ortho*-quinones [31]. Degassing of the capture buffers is also required.

*pH:* Polyphenolic compounds are prone to autoxidation, which can lead to uncontrolled activation of the probes and hence probably rapid and non-selective capture of the most abundant proteins. Autoxidation of the polyphenolic probes can be minimized by degassing all buffers and maintaining a slight acidity (pH 6.8). However, the desired NaIO_4_-mediated oxidation of the polyphenolic probes during the activation step would be more favorable and selective at a neutral to slightly alkaline pH [31]. A slight increase in pH is ensured through the use of a dilute solution of NaIO_4_ in PBS pH 7.4. The final addition of the quenching (reducing) sodium dithionite (Na_2_S_2_O_4_) reagent generates acids (e.g., sulfurous acid), which lower the pH to minimize any further autoxidation.

*Titration of periodate:* We have developed a titration protocol for determining the minimal amount of NaIO_4_ necessary to activate the probes. Avoiding the addition of excess NaIO_4_ by determining the exact amount required for the proteome under investigation can reduce the risk of undesired side reactions (vide infra). The titration protocol relies on the use of 2,6-dichlorophenolindophenol (DCPIP) as an indicator [23,24] and further enables observation of the quenching process (see the Appendix A for details). Usually, a 10 mM periodate solution is sufficient, but if ascorbate is present in the protein buffer, which is the case here, a 100 mM periodate solution is used. 

*Periodate-mediated probe activation:* This oxidation step leading to the formation of probe-derived *ortho*-quinones occurs at the second timescale [31]. Keeping the oxidation time to just about one minute can help to avoid unwanted side reactions, such as the oxidative cleavage of 1,2-diols or 1,2-amino alcohols, which usually requires several minutes of reaction time. Longer exposure of proteins to NaIO_4_ may lead, for example, to undesired protein-protein cross-linking through the oxidative cleavage of *N*-terminal serine or threonine residues (i.e., 1,2-amino alcohols) [25].

*Quenching of reactive species*: Instead of the traditionally used organic reducing agent DTT, which can generate reactive aldehydes upon oxidative cleavage of its 1,2-diol unit by NaIO_4_, we used the inorganic sodium dithionite (Na_2_S_2_O_4_) to inactivate any excess of NaIO_4_ and to reduce any remaining probe-derived *ortho*-quinones [32,33]. A four-fold excess of Na_2_S_2_O_4_ relative to the amount of NaIO_4_ is usually sufficient.

*Protein Pull-down with beads:* Usually, 50 μL of the commercial suspension of beads is used, which corresponds to a binding capacity of 200 pmol of biotinylated proteins. Tween (0.1% final concentration) is used to avoid unspecific hydrophobic interactions between membrane proteins.

*Washing buffers:* DTT was also replaced by β-mercaptoethanol in the washing buffers in order to avoid adding several different reagents, β-mercaptoethanol being present in the Laemmli buffer used for elution of the beads (i.e., recovery of captured proteins, vide infra). The washing procedure included a step using 0.2% SDS for 10 min, while shaking [34], which resulted in much cleaner negative controls and an increase in the quality of the captures.

*Capture assays*: Capture assays were performed using flavanol-bearing probes **Q8**, **Q6,** or **K8** (2 μL of 200 μM in DMSO, 400 pmol) with suspensions of microsomal petal protein (for *Rudbeckia:* 99 μL/for *Tagetes:* 60 μL). For the negative controls, 2 μL of DMSO were added instead of the probe. Each sample (probe-containing and negative controls) was prepared in triplicate. Incubations with proteins were carried out in 1.5 mL Eppendorf low-binding tubes under shaking (1150 rpm) at 37 °C for 30 min. Then, the amount of NaIO_4_, as determined by our DCPIP titration method (see the Appendix A), was added to the protein samples: for *Rudbeckia*, 20 µL NaIO_4_ (100 mM) was diluted into 170 µL PBS pH 7.4, and for *Tagetes*, 32 µL NaIO_4_ (100 mM) was diluted into 178 µL PBS pH 7.4. The mixtures were allowed to shake (1150 rpm) at 23 °C for 1 min and then quenched by the addition of 400 μL of quenching buffer (4 mM Na_2_S_2_O_4_ in PBS pH 6.8). Samples were shaken again (1150 rpm) for 2 min and subsequently incubated with streptavidin-coated magnetic beads (50 μL, Dynabeads^®^MyOneTM Streptavidin C1: prewashed and suspended in 250 μL of PBS pH 6.8 containing Tween, 0.1% final concentration) at 23 °C for 1 h under shaking (1150 rpm). The beads were collected using a magnetic device, and the supernatants were removed. The beads were first washed three times with some washing buffer (1 mL × 3 [(50 mM Tris-HCl pH 6.8, 0.05% (wt/vol) octyl-β-D-glucopyranoside, 0.5 M NaCl, 2 mM β-mercaptoethanol)]), then twice with an aqueous solution of SDS 0.1% (1 mL × 2), once with a solution of SDS 0.2% + 2 mM β-mercaptoethanol in PBS buffer pH 6.8 (1 mL, 10 min at 23 °C, 1150 rpm), then once with MilliQ water (1 mL). At this stage, the Eppendorf tubes were changed (new low-binding tubes) and the beads were washed with PBS buffer pH 6.8 (1 mL). After removal of the last supernatant, a denaturing Laemmli buffer (30 μL, containing SDS and β-mercaptoethanol; see the Appendix A) was then added to the beads, and the resulting suspensions were heated at 95 °C for 5 min under shaking (1400 rpm) in order to release the captured proteins from the beads. The resulting probe-labeled proteins were then analyzed by label-free quantitative shotgun proteomics (see infra and also Appendix A).

This protocol was also implemented using probe **Q8** without the addition of NaIO_4_ (Method **N**) with the aim of assessing the extent of non-covalent captures and/or possible covalent captures through autoxidation, a process that can never be totally excluded when working with polyphenolic materials.

### 3.3. Label-Free Quantitative Shotgun Proteomics 

*Sample preparation and protein digestion*: Protein samples were solubilized in Laemmli buffer and deposited onto SDS-PAGE gels for concentration and cleaning purposes. Separation was stopped once proteins entered the resolving gel. After colloidal blue staining, bands were cut out of the SDS-PAGE gel and subsequently cut into 1 mm × 1 mm gel pieces. Gel pieces were destained in 25 mM ammonium bicarbonate and 50% acetonitrile (can), rinsed twice in ultrapure water, and shrunk in ACN for 10 min. After ACN removal, gel pieces were dried at room temperature, covered with the trypsin solution (10 ng/µL in 50 mM NH_4_HCO_3_), rehydrated at 4 °C for 10 min, and finally incubated overnight at 37 °C. Spots were then incubated for 15 min in 50 mM NH_4_HCO_3_ at room temperature with rotary shaking. The supernatant was collected, and an H_2_O/ACN/HCOOH (47.5:47.5:5) extraction solution was added to gel slices for 15 min. The extraction step was repeated twice. The supernatants were pooled and dried in a vacuum centrifuge. Digests were finally solubilized in 0.1% HCOOH.

*nLC-MS/MS Analysis and label-free quantitative data analysis*: Peptide digests were analyzed on an Ultimate 3000 nanoLC system (Dionex, Amsterdam, The Netherlands) coupled to an Electrospray Orbitrap Fusion™ Lumos™ Tribrid™ Mass Spectrometer (Thermo Fisher Scientific, San Jose, CA, USA). Thus, peptide digests (10 µL) were loaded onto a 300-µm-inner diameter × 5-mm C_18_ PepMap^TM^ trap column (LC Packings, Thermo Fisher Scientific, San Jose, CA, USA) at a flow rate of 10 µL/min. The peptides were eluted from the trap column onto an analytical 75-mm id × 50-cm C18 Pep-Map column (LC Packings) with a 4–40% linear gradient of solvent B in 45 min (solvent A was 0.1% formic acid and solvent B was 0.1% formic acid in 80% ACN). The separation flow rate was set at 300 nL/min. The mass spectrometer operated in positive ion mode at a 2-kV needle voltage. Data were acquired using Xcalibur 4.3 software in a data-dependent mode. MS scans (*m*/*z* 375–1500) were recorded in the Orbitrap at a resolution of R = 120,000 (@ *m*/*z* 200) and an AGC target of 4 × 10^5^ ions was collected within 50 ms. Dynamic exclusion was set to 60 s and top speed fragmentation in HCD (Higher-energy Collisional Dissociation) mode was performed over a 3 s cycle. MS/MS scans were collected in the ion trap with a dynamic maximum injection time. Only +2 to +7 charged ions were selected for fragmentation. Other settings were as follows: no sheath nor auxiliary gas flow; the heated capillary temperature of 275 °C; normalized HCD collision energy of 35%; isolation width of 1.6 m/z; AGC target of 2 × 10^3^; and normalized AGC target of 20%. Monoisotopic precursor selection (MIPS) was set to peptide, and an intensity threshold was set to 5 × 10^3^.

*Database search and results processing*: Data were searched by SEQUEST through Proteome Discoverer 2.4 (Thermo Fisher Scientific Inc.) against different databases. For *Rudbeckia hirta* protein samples (total microsome and capture samples), two distinct searches were performed against the *Helianthus annuus* UNIPROT database (RPS_2019_09_Helianthus_annuus_UP000215914.fasta) supplemented by F6H sequences from *Bidens ferulifolia* (NCBI MT882688, MT882689) and from *Rudbeckia hirta* (NCBI MT875175) or against the *Rudbeckia hirta* proteomic database (New_rudbeckia_rename.fasta) derived from the combination of SAMN14886687 and SAMN14886688 transcriptomic databases provided by the Halbwirth laboratory, identically supplemented by F6H sequences from *Bidens ferulifolia* (NCBI MT882688, MT882689) and from *Rudbeckia hirta* (NCBI MT875175). For *Tagetes erecta* protein samples (total microsome and capture samples), two distinct searches were performed against the *Helianthus annuus* UNIPROT database (RPS_2019_09_Helianthus_annuus_UP000215914.fasta) supplemented by F6H sequence from *Rudbeckia hirta* (NCBI MT875175) and against the *Tagetes erecta* proteomic database (GGGQ01.1_proteomic.fasta) derived from the transcriptomic database [27] and supplemented by F6H sequence from *Tagetes erecta* (NCBI MT882687). Spectra from peptides higher than 5000 Da or lower than 350 Da were rejected. A precursor detector node was included. Search parameters were as follows: The mass accuracy of the monoisotopic peptide precursor and peptide fragments was set to 10 ppm and 0.6 Da, respectively. Only b- and y-ions were considered for mass calculation. Oxidation of methionines (+16 Da), methionine loss (−131 Da), methionine loss with acetylation (−89 Da), and protein *N*-terminal acetylation (+42 Da) were considered variable modifications, while carbamidomethylation of cysteines (+57 Da) was considered a fixed modification. Two missed trypsin cleavages were allowed. Peptide validation was performed using the Percolator algorithm [35], and only “high confidence” peptides were retained, corresponding to a 1% false-positive rate at the peptide level. Peaks were detected and integrated using the Minora algorithm embedded in the proteome discoverer. Protein abundances were calculated as the sum of unique peptide intensities. Normalization was performed based on total protein amounts. Protein ratios were calculated from the grouped protein abundances. An ANOVA test was calculated based on individual protein values. Quantitative data were considered for proteins quantified by a minimum of two peptides and a statistical *p*-value lower than 0.05.

### 3.4. Bioinformatics

After proteomic analysis of all capture and negative control samples, bioinformatic processing by in-house Python scripts [36] of the resulting raw data identified, for each assay, statistically significant (*p*-values < 0.05) captured proteins from proteins detected in the total proteomes (see the Appendix A for details). Highly significant proteins were defined as those proteins that were either not present in the negative control or strongly enriched in the capture sample; strong enrichment was statistically defined according to the following criteria: (log2(r) > 2 and –log(*p*) > 2.5); *p*-value: probability for a differential abundance of a protein caused by the probe effect; r-value: fold change of each protein in a probe-containing sample as compared to the probe-less negative control.

A gene ontology (GO) enrichment analysis was also performed on all proteins significantly captured by probe **Q8** from *Rudbeckia* microsomes in order to further qualify the functions of these proteins, thus revealing a highly significant enrichment of proteins involved in catalytic activities, including catalytic functions related to oxidoreductase activity, in agreement with our search for F6H and F8H proteins (see the Appendix A, for details).

A tailored BlastP search [37] was used to query the proteome of *Rudbeckia hirta* (Asteraceae). The query file consisted of the F6H protein sequences of *Rudbeckia hirta* as well as the F6H and F8H protein sequences of *Scutellaria baicalensis* (Lamiaceae) (see the Appendix A for details).

## 4. Conclusions

We have developed efficient and unveiling methods for the chemical synthesis of biotinylated C6- and C8-linked flavonol-bearing probes and for the chemoproteomic capture of proteins involved in the biosynthesis of flavonol anabolites. Two novel flavonol 6-hydroxylases (F6H) recently identified by means of molecular biology from *Tagetes* and *Rudbeckia* species were thus captured, as well as several candidates for the identification of other flavonoid-specific enzymes acting as oxidases, such as flavonol 8-hydroxylases (F8H), or as 6-*O*- and 8-*O*-methyl- or glycosyltransferases. This approach, which combines the design and synthesis of molecular probes, their use in affinity enrichment chemoproteomics, and the analysis of capture results by bioinformatic tools, constitutes a powerful methodology for substrate-based pinpointing of known and yet unknown flavonoid enzymes from plant protein extracts. The data thus gathered will guide and should expedite further studies aimed at fully identifying and characterizing novel enzymes and their substrate specificity through more classical molecular biology and enzymology experiments toward a more comprehensive and full understanding of flavonoid biosynthesis.

## Figures and Tables

**Figure 1 ijms-24-09724-f001:**
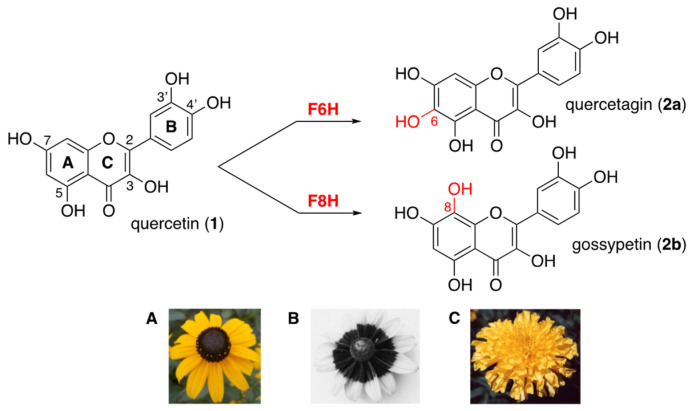
Higher hydroxylation of quercetin (**1**) into quercetagin (**2a**) or gossypetin (**2b**) through the action of flavonol 6- and flavonol 8-hydroxylases (**F6H** and **F8H**); (**A**) *Rudbeckia hirta*; (**B**) UV photography of *Rudbeckia hirta* showing the circular dark honey guide rich in flavonols bearing an additional hydroxyl group at position 6 or 8 (bull’s eye effect); (**C**) *Tagetes erecta*.

**Figure 2 ijms-24-09724-f002:**
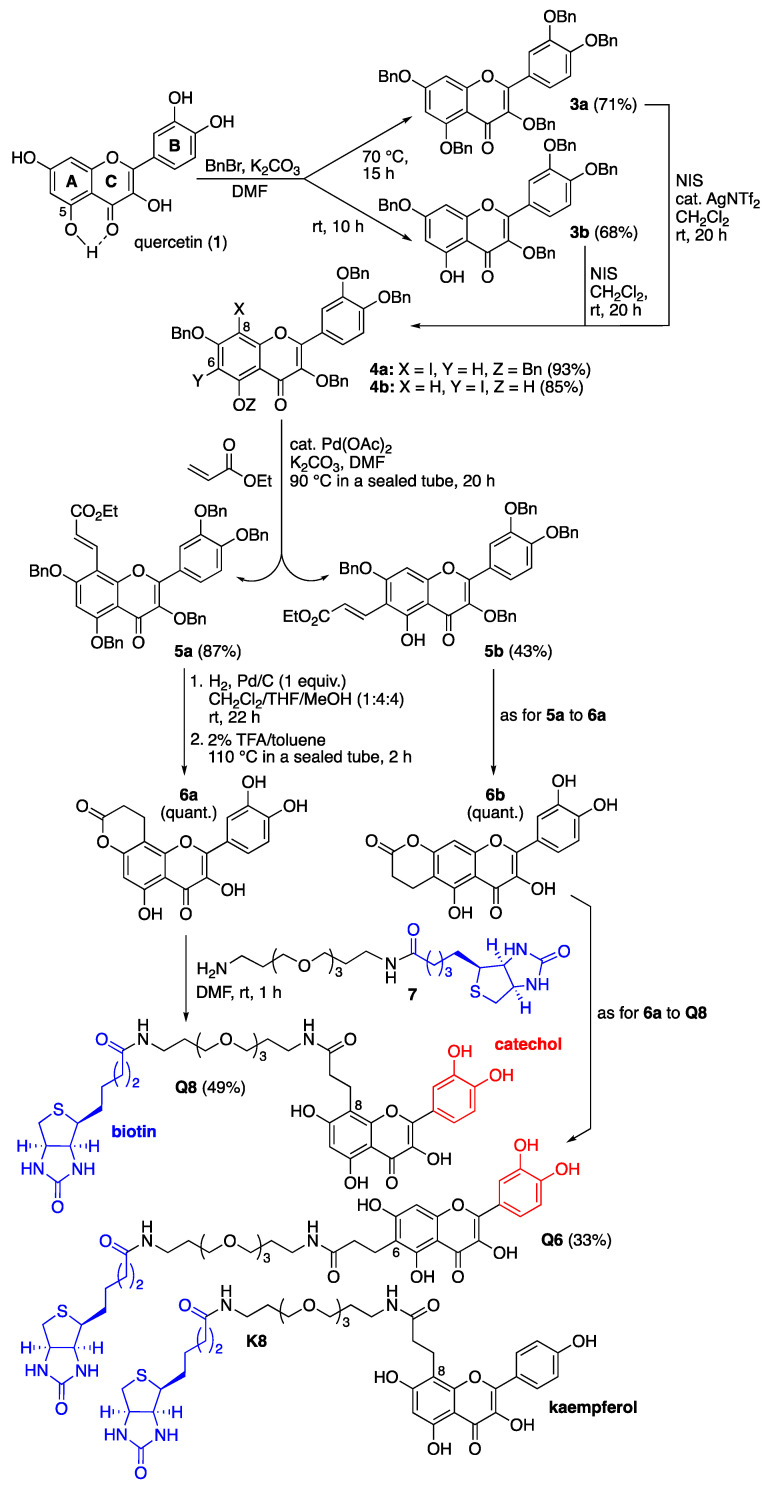
Chemical synthesis of C8- and C6-linked quercetin-bearing probes **Q8** and **Q6**, and structure of the C8-linked kaempferol-bearing probe **K8**. Details of the synthesis of these probes and their characterization data are given in the Appendix A.

**Figure 3 ijms-24-09724-f003:**
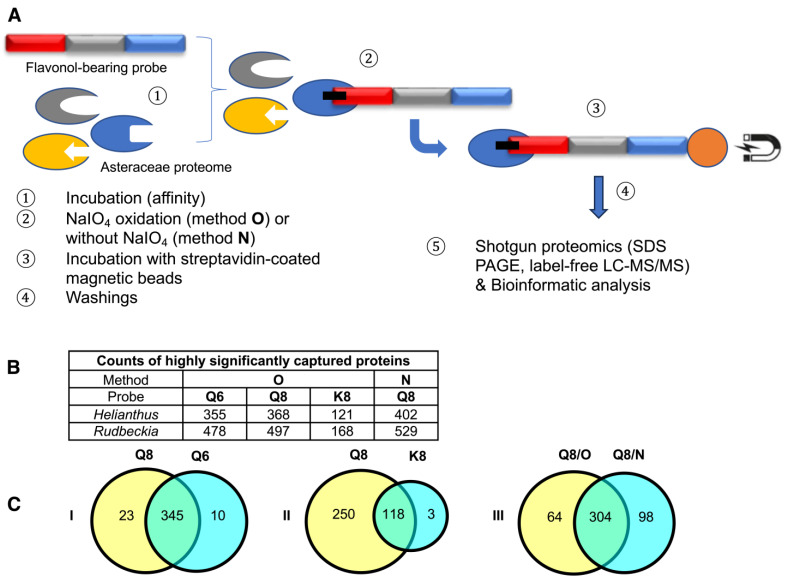
(**A**) Schematic representation of the affinity-based chemoproteomic capture process. (**B**) Table showing the counts of highly significant proteins captured from *Rudbeckia* microsomes through methods **O** (with NaIO_4_-mediated probe activation) and **N** (without using NaIO_4_). Interrogation: *Helianthus* proteome data and *Rudbeckia* transcriptome data. (**C**) Venn diagrams showing the influence of the probes (**Q6**, **Q8**, and **K8**) or the methods (**O** and **N**) on the counts of captured proteins (*Helianthus* proteome data). For the definition of highly significant proteins, see Figure 4 and the Section 3. The full lists of all highly significant proteins and of those captured from two assays (intersections) are available in the Appendix A; see also Appendix A for the counts of total proteins and all significantly captured proteins.

**Figure 4 ijms-24-09724-f004:**
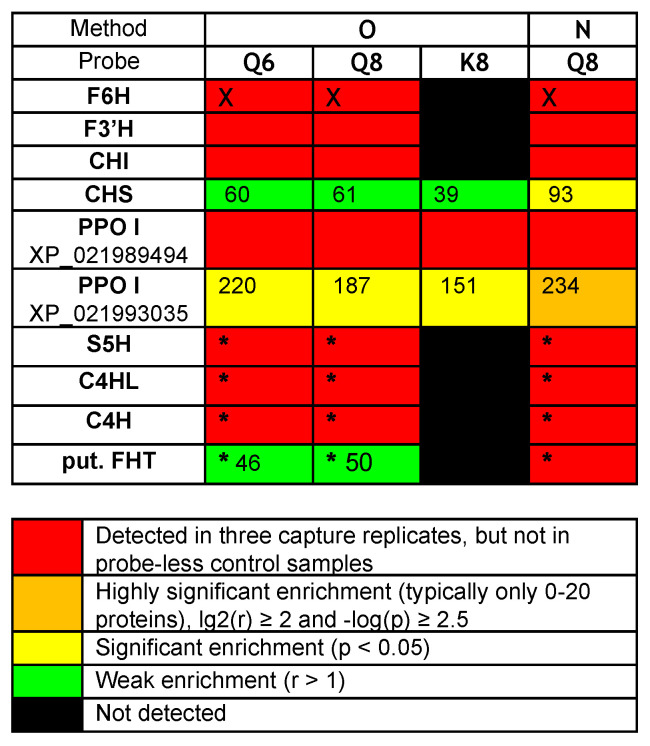
Heatmap of relevant protein captures (i.e., known flavonoid enzymes and aromatic hydroxylases/oxidases) from *Rudbeckia hirta* petal microsomes: F6H (the *Rudbeckia* flavonol-6-hydroxylase), F3′H (flavonoid 3’-monooxygenase-like isoform X1), CHI (chalcone-flavanone isomerase), CHS (chalcone synthase), PPO I XP_021989494 (polyphenol oxidase I, chloroplastic-like), PPO I XP_021993035 (polyphenol oxidase I, chloroplastic-like, pigment biosynthetic process), S5H (salicylate 5-hydroxylase), C4HL (trans-cinnamate 4-monooxygenase-like), C4H (trans-cinnamate 4-monooxygenase), put. FHT (a putative flavanone 3-dioxygenase). Interrogation: (i) *Helianthus* proteome data, (ii) supplemented with the *Rudbeckia* F6H sequence (marked with X), and (iii) *Rudbeckia* transcriptome data (marked with *). *p*-value: probability for differential abundance of a protein caused by the probe effect; r-value: the fold change (“enrichment”) of the measured abundance of each protein in a probe-containing sample as compared to the probe-less negative control—the numbers in this heatmap indicate the respective fold change for each protein.

**Figure 5 ijms-24-09724-f005:**
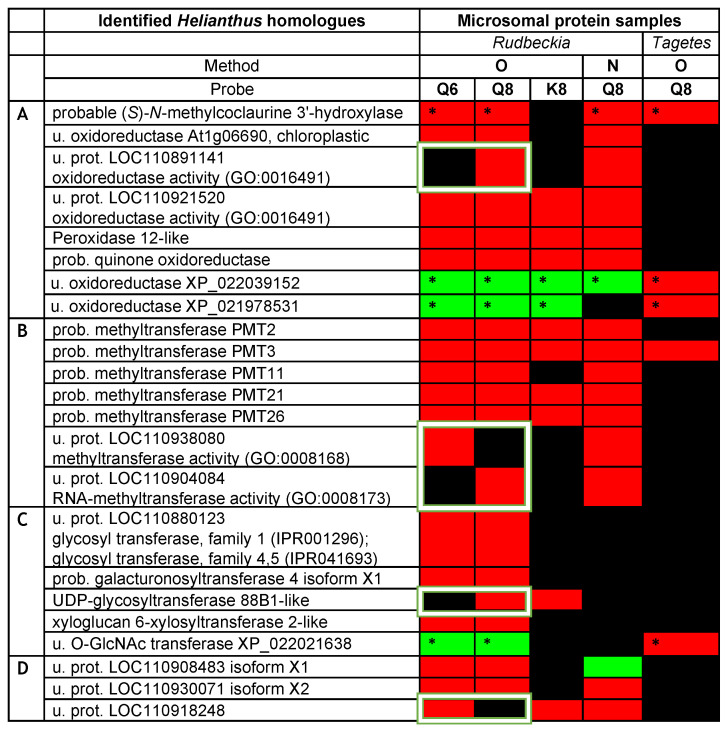
Heatmap of potentially relevant uncharacterized or not fully characterized proteins as candidates for oxidases (**A**), methyltransferases (**B**), or glycosyltransferases (**C**) captured from *Rudbeckia* and *Tagetes* petal microsomes. Interrogation: *Helianthus* proteome data and *Rudbeckia or Tagetes* transcriptome data (marked with *); u. = uncharacterized; prot. = protein; prob. = probable. The white frames highlight only one of the isomeric quercetin-bearing probes, **Q6** or **Q8**. Proteins in category (**D**) are not annotated or do not belong to categories (**A**–**C**). The color coding of the detection level is the same as that used in Figure 4. The F6H proteins are annotated as their *Helianthus* homologue “probable (*S*)-*N*-methylcoclaurine 3’-hydroxylase”.

**Figure 6 ijms-24-09724-f006:**
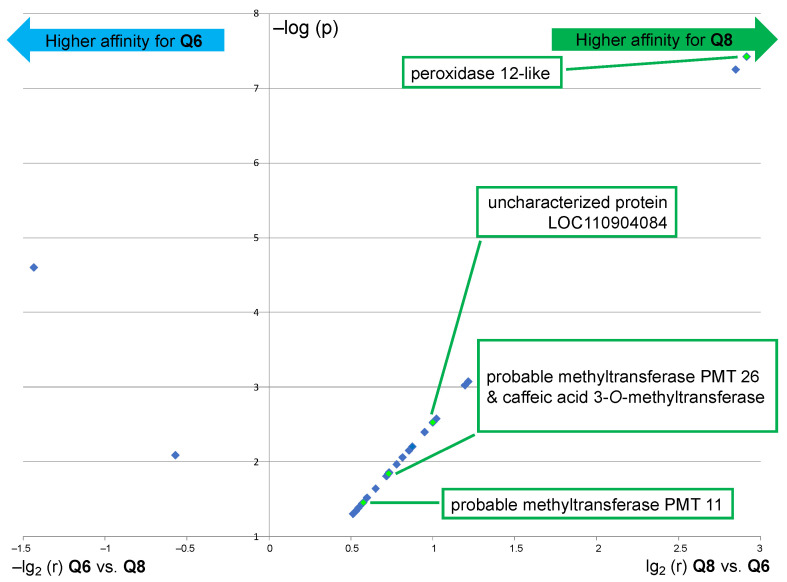
Volcano plot showing the significantly enriched proteins (*p* < 0.05) in the capture sample using probe **Q8** as compared to the capture sample using probe **Q6** (right part) and vice versa (left part). Interrogation: *Helianthus* proteome data (similar results were obtained through interrogation of the Rudbeckia transcriptome data). NB1: only highly significantly captured proteins (red color coding in the heatmap of Figure 5, i.e., not present in negative controls) are displayed; NB2: only proteins with biochemical relevance to the present study are named; see also the Appendix A, for the list of all significantly enriched proteins.

**Figure 7 ijms-24-09724-f007:**
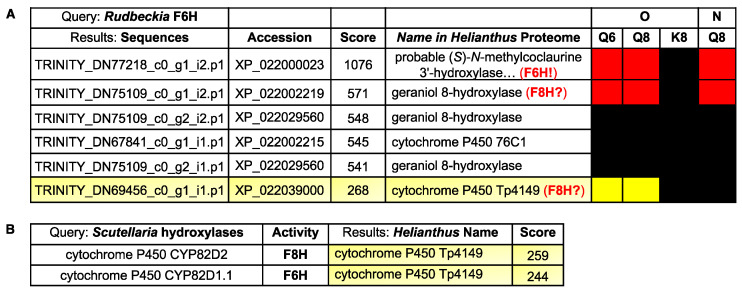
(**A**) BlastP results of the comparison of the *Rudbeckia* F6H sequence with the *Rudbeckia* transcriptome data. From a total of 30 sequences producing significant alignments, the two proteins with the highest similarity scores (>500) and one other protein with a lower score (<300) were captured by our probes **Q6** and **Q8**, as indicated in the heatmap on the right (see Figure 4 for color coding of levels of detection significance). Accession: protein identifyr in the NCBI database. (**B**) BlastP results of the comparison of two flavonol hydroxylase sequences present in *Scutellaria baicalensis* with the *Rudbeckia* transcriptome data.

## Data Availability

The data that support this study are available from the corresponding authors upon request. The ^1^H and ^13^C spectra of the probes **Q6**, **Q8**, and **K8**, as well as those of the advanced intermediates of their synthesis, are included in the Appendix A. The mass spectrometry proteomics data have been deposited to the ProteomeXchange Consortium via the PRIDE [38] partner repository with the dataset identifier PXD041520 (username: reviewer_pxd041520@ebi.ac.uk; password: YUuPGMal).

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
