# Peer review of "Synthesis of Flavonol-Bearing Probes for Chemoproteomic and Bioinformatic Analyses of Asteraceae Petals in Search of Novel Flavonoid Enzymes"

_ijms, 2023, doi:10.3390/ijms24119724_

Round 1

Reviewer 1 Report

The authors wish to report the chemical synthesis of few flavonol-bearing probes and their application for the chemoproteomic capture of proteins involved in the biosynthesis of flavonol anabolites. The related design and synthesis of molecular probes, their use in affinity enrichment chemoproteomics and the analysis of capture results by bioinformatic tools establishes a powerful methodology for substrate-based pinpointing of known and yet unknown flavonoid enzymes from plant protein extracts.

Overall, the work and the obtained results have the sufficient significance for publication in this journal. The manuscript has also been prepared properly. To further improve the quality of the manuscript and for easy understanding to the general readers, the author may perform a minor revision by including a schematic diagram of the overall work performed in this manuscript.

Reviewer 2 Report

Kempf et al designed and synthesized two flavonol-bearing probes for the pull-down and chemoproteomic study of novel flavonoid enzymes. The authors identified hundreds of proteins with the two probes in two species. Known and novel flavonoid enzymes are found among the identified proteins. The probes are efficient but they apparently are not specific enough to only pull-down flavonoid enzymes. Also, the authors used an crosslinking strategy but it turned out that the probe can be used without oxidation crosslinking, so the protocol can be optimized. 

1.     It is surprising that Q8 O and Q8 N conditions both pulled down large number of proteins, with many shared ones. A control with reducing reagents would show if it was due to high affinity or endogenous oxidants.

2.     Among the hundreds of proteins found by the probes, the authors mentioned several types of flavonoid-specific enzymes. What are the functions of the rest of the proteins. Do they also bind to the probes or this is just non-specific binding?
